# The role of memory in non-genetic inheritance and its impact on cancer treatment resistance

**Tyler Cassidy**[1]*, **Daniel Nichol**[2], **Mark Robertson-Tessi**[3], **Morgan Craig**[4,5], **Alexander R. A. Anderson**[3]*

**1** Theoretical Biology and Biophysics, Los Alamos National Laboratory, Los Alamos, New Mexico, United States of America, **2** Evolutionary Genomics and Modelling Lab, Centre for Evolution and Cancer, The Institute of Cancer Research, London, United Kingdom, **3** Department of Integrated Mathematical Oncology, H. Lee Moffitt Cancer Center, Tampa, Florida, United States of America, **4** Département de mathématiques et de statistique, Université de Montréal, Montreal, Canada, **5** CHU Sainte-Justine, Montreal, Canada

* tyler.cassidy@mail.mcgill.ca (TC); Alexander.Anderson@moffitt.org (ARAA)

**Data Availability Statement:** The data underlying the results presented in the study are available in the supplementary material of Craig et al. (2019), DOI: 10.1371/journal.pcbi.1007278.

## Abstract

Intra-tumour heterogeneity is a leading cause of treatment failure and disease progression in cancer. While genetic mutations have long been accepted as a primary mechanism of generating this heterogeneity, the role of phenotypic plasticity is becoming increasingly apparent as a driver of intra-tumour heterogeneity. Consequently, understanding the role of this plasticity in treatment resistance and failure is a key component of improving cancer therapy. We develop a mathematical model of stochastic phenotype switching that tracks the evolution of drug-sensitive and drug-tolerant subpopulations to clarify the role of phenotype switching on population growth rates and tumour persistence. By including cytotoxic therapy in the model, we show that, depending on the strategy of the drug-tolerant subpopulation, stochastic phenotype switching can lead to either transient or permanent drug resistance. We study the role of phenotypic heterogeneity in a drug-resistant, genetically homogeneous population of non-small cell lung cancer cells to derive a rational treatment schedule that drives population extinction and avoids competitive release of the drug-tolerant sub-population. This model-informed therapeutic schedule results in increased treatment efficacy when compared against periodic therapy, and, most importantly, sustained tumour decay without the development of resistance.

## Author summary

We propose a simple mathematical model to understand the role of phenotypic plasticity and non-genetic inheritance in driving therapy resistance in cancer. We identify the role of non-genetic inheritance on treatment resistance and use a variety of analytical and numerical techniques to understand the impact of phenotypic plasticity on population fitness and dynamics. We further use our model to study the role of phenotypic heterogeneity in therapeutic resistance in a genetically identical non-small cell lung cancer population. Finally, we combine analytical perspectives and techniques from the theory of

**Funding:** TC was partially supported by the Natural Sciences and Research Council of Canada (NSERC) through the PGS-D program and NIH grants R01-AI116868 and R01-OD011095. Portions of this work were performed under the auspices of the U.S. Department of Energy under contract 89233218CNA000001. DN received no specific funding for this work. MC was funded by NSERC Discovery grant and Discovery Launch Supplement RGPIN-2018-04546. MRT and ARAA were funded by the Cancer Systems Biology Consortium and the Physical Sciences Oncology Network at the National Cancer Institute, through grants U01CA232382 and U54CA193489. Support from the Moffitt Center of Excellence for Evolutionary Therapy. The funders had no role in study design, data collection and analysis, decision to publish, or preparation of the manuscript.

**Competing interests:** The authors have declared that no competing interests exist.

structured populations, renewal equations, and infinite dimensional dynamical systems to derive a model-informed therapeutic strategy that both drives tumour eradication and avoids competitive release of a drug-tolerant subpopulation. These results exemplify the potential of using mathematical techniques to identify therapeutic strategies to guide the evolution of a heterogeneous tumour.

## Introduction

Intra-tumour heterogeneity is a leading driver of cancer treatment failure [1–3]. The genetic instability and high proliferative capacity typical of cancer cells induces a genetically heterogenous population in which resistance-conferring mutations can arise and expand during the selective pressure of therapy. This evolutionary process leads to the eventual failure of treatment and the outgrowth of a refractory tumour [3–9]. However, it is increasingly understood that genetic aberrations are not the sole mechanism through which drug-resistant phenotypes can arise. Rather, adaptive phenotypic changes can arise without an associated genetic mutation. Such phenotypic heterogeneity has been extensively studied as a possible mechanism of treatment resistance [7, 8, 10–14]. For example, chemotherapy has been shown to induce a transient drug-tolerant phenotype in breast cancer cell lines such that re-sensitisation occurs following cessation of therapy [15, 16], an example of phenotypic plasticity [17]. Whilst this heterogeneity arises in response to environmental change, non-genetic variation in phenotypes can also arise in unchanged environments, indicating the presence of stochastic phenotype switching, termed bet-hedging [13, 14]. Bet-hedging induces phenotypic diversity that can help protect a population from extinction following catastrophic environmental changes such as cytotoxic therapy [18–21].

In recent years, evolutionarily-informed cancer therapy regimens have arisen as a potential strategy to delay the emergence of drug resistance. *Adaptive therapies* exploit competition between clonal populations by incorporating periods without therapy wherein resistant subclones, which are often assumed to have a fitness cost in the absence of treatment, can be outcompeted by drug-sensitive clones [5, 22–24]. The treatment is applied and removed based on one or more biomarkers of disease, typically proxy measurements for tumour burden. The theory underlying cancer adaptive therapy is primarily based on competition dynamics between tumour subclones that are not plastic, for example clones arising from genetic mutations. It is presently unclear whether adaptive therapies will prove as effective in mitigating resistance driven by non-genetic mechanisms that change on a faster timescale than mutational rates, or whether better understanding of such non-genetic drivers of resistance could help in the design of more effective evolutionary therapies. Here, we address this question and study the impact of bet-hedging strategies on the development of treatment resistance by developing a simple and qualitative mathematical model.

Mathematical models have been used extensively to understand the development of resistance to anti-cancer therapies. A number of authors have considered how phenotypic variation arises [25, 26] as well as the effects of phenotypic heterogeneity (see [11] and references therein). Recent modelling efforts used gene-regulatory networks or branching-type formulations to investigate the role of phenotypic switching on treatment resistance [27, 28], while other authors have used Markov processes to illuminate the role of stochastic phenotypic switching in treatment resistance [29–31]. In addition to these approaches, many models rely, in large part, on the use of structured equations which bridge cellular dynamics and population level heterogeneity by explicitly considering the cellular phenotype. Often formulated as partial

differential equations (PDEs) structured in phenotypic space, these models conceptualise continuously varying cellular phenotypes [32–35]. These PDE models often include non-local terms to incorporate interactions between cells of different phenotypes, and solutions of these models typically describe the density of cells in phenotype space. Consequently, these structured mathematical models are well-suited to study phenotypic evolution in dynamic environments. Here, we develop a simple structured PDE model to study the role of phenotype plasticity in the evolution treatment resistance.

We are particularly interested in the role of stochastic phenotype switching on the development of resistance to anti-cancer therapies. It has previously been shown that stochastic switching between quiescence and proliferation in mammalian cells is biased by the inheritance of mitogen and p53 signalling factors at cell division [36]. The concentration of these factors was shown to be dependent on the life history of the parental cell and are thus representative of non-genetic 'memory' in phenotypic switching [36]. Theoretical studies of chemical reaction networks have demonstrated that simple combinations of catalytic and autocatalytic reactions can produce such bistable switches, with different network structures inducing different convergence and stability behaviour [37, 38]. Coupled with the inheritance and subsequent decay of intracellular signalling factors, these bistable switches can govern a diverse range of memory-driven switching regimes. Here, we investigate the role of phenotypic memory in intracellular inheritance in driving the emergence of a resistant phenotype during therapy. To this end, we use the cellular age, rather than phenotypic state, to structure our model. In this framework, the cellular age acts as a cipher for a number of epigenetic factors that vary throughout the cell's life, such as protein accumulation/dilution, cell size, adaptation to environmental stresses, etc. In the model, we use the age of the parent cell to determine the probability that daughter cells will inherit the parental phenotype: cells that reproduce soon after their birth are more likely to bequeath their phenotype to their offspring. This mapping from cellular age to switching probability generalises the role of the bistable switch mechanism that governs phenotypic differentiation, as well any age-driven changes to its behaviour.

The canonical example of the stochastic nature of phenotypic inheritance is the existence of persister cells resulting from stochastic phenotypic switches in *Escherichia coli* populations [14, 20, 39, 40]. Comparatively rare in a population in stable exponential growth, the proportion of persister cells increases as the population of *E.coli* cells competes for limited resources [20, 39]. Accordingly, we use our model to study the role of growth phase on population composition. Specifically, we demonstrate that populations in exponential and stationary growth stages respond differently to environmental changes. For example, we show that changes in the relative fitness between two phenotypically distinct populations has a drastically different impact on a population in exponential growth compared against the same change during the stationary growth phase. Further, we study the role of phenotypic cooperation on population growth in nutrient-rich environments, and show that this cooperation can hasten population growth when compared against purely Malthusian growth.

We also study the establishment of a resistant population during cytotoxic treatment. Through numerical simulation, we demonstrate that different phenotype switching strategies result in either transient resistance [15, 16], or permanent resistance due to the establishment of a dominant, resistant population [41, 42]. We then investigate alternative treatment scheduling options to delay or avoid the establishment of resistance by preserving a drug-sensitive population. This scheduling, inspired by adaptive therapy [5, 22], is then shown to outperform periodic or maximally tolerable dosing strategies in a result that is robust to parameter changes. Applying our model to *in vitro* growth data from genetically homogeneous non-small cell lung cancer (NSCLC) populations, we study the effect of phenotypic switching in resistance to anti-cancer drugs. We use two different measurements of population fitness under

treatment to derive a model-informed therapeutic schedule that balances the desire to drive tumour extinction with the need to avoid competitive release of a drug-tolerant population and the resulting therapy resistance. We show that this treatment schedule leads to long term tumour decay and significantly outperforms metronomic dosing. In the interest of clarity, we present the full analytic results in S1 Text.

## Results

### Phenotypic switching model

Our primary interest is to understand and quantify resistance during chemotherapy. For this, consistent with previous experimental [15, 16, 39, 43] and theoretical [7, 13, 21, 22, 42, 44] studies of bet-hedging, we constructed a mathematical model of phenotypic switching to track the density of cells with a drug-sensitive ($A(t, a)$) or drug-tolerant ($B(t, a)$) phenotype at time $t$ and age $a$. The object of clinical interest at time $t$ is unlikely to be the density of cells with a given age, but rather total number of cells of each phenotype, given by

$$\bar{A}(t) = \int_0^\infty A(t, a) \mathrm{d}a \quad \text{and} \quad \bar{B}(t) = \int_0^\infty B(t, a) \mathrm{d}a. \tag{1}$$

In what follows, the total number of cells is denoted by $N(t) = \bar{A}(t) + \bar{B}(t)$.

We assume that cell phenotypes are fixed at birth [13] and reproduce at rates $R_A(\bar{A}(t), \bar{B}(t))$ and $R_B(\bar{A}(t), \bar{B}(t))$, respectively. Briefly, we consider multiple forms of $R_A(\bar{A}(t), \bar{B}(t))$ and $R_B(\bar{A}(t), \bar{B}(t))$ corresponding to different biological assumptions. When considering populations with (effectively) unlimited resources, such as those that are continually replated during *in vitro* experiments, we use a Malthusian growth model with $R_A(\bar{A}(t), \bar{B}(t)) = r_A$ and $R_B(\bar{A}(t), \bar{B}(t)) = r_B$. We also consider the resource limited case, such as *in vitro* experiments that approach total confluence, and use a logistic growth models for $R_A(\bar{A}(t), \bar{B}(t))$ and $R_B(\bar{A}(t), \bar{B}(t))$. Finally, we incorporate the effects of phenotypic cooperation, whereby a larger proportion of cells of a certain phenotype can lead to increased phenotypic expansion through an Allee effect or frequency dependent fitness changes through a function $f_n(\bar{A}(t), \bar{B}(t))$ [2, 45–48]. The function $f_n(\bar{A}(t), \bar{B}(t))$ models the increase in relative fitness of drug-tolerant cells as they become more common and determining a precise formulation for $f_n(\bar{A}(t), \bar{B}(t))$ is difficult [23]. We use a Hill function formulation of $f_n(\bar{A}(t), \bar{B}(t))$ with Hill coefficient $n$ that modulates the type of Allee effect. We give the functional forms of $R_A$ and $R_B$ in each case in section *Growth dynamics* in S1 Text.

Finally, we assume that drug-sensitive and drug-tolerant cells have phenotype-specific death rates $d_A$ and $d_B$. Under these assumptions, $A(t, a)$ and $B(t, a)$ satisfy the age structured PDE,

$$\left. \begin{array}{rl} \partial_t A(t, a) + \partial_a A(t, a) &= -[d_A + R_A(\bar{A}(t), \bar{B}(t))]A(t, a) \\ \partial_t B(t, a) + \partial_a B(t, a) &= -[d_B + R_B(\bar{A}(t), \bar{B}(t))]B(t, a) \end{array} \right\} \tag{2}$$

In this way, we studied the cellular ageing process over (linear) time (LHS Eq (2)), with cellular loss at age $a$ due to either death or reproduction (RHS Eq (2)). As dividing cells necessarily have age $a > 0$, cellular reproduction results in the the removal of the mother cell, which accounts for the negative sign on the RHS of (2). These reproducing cells produce two daughter cells with age $a = 0$ that re-enter the model through the boundary conditions for $A(t, 0)$ and $B(t, 0)$. Accordingly, we model cellular reproduction through the non-local boundary conditions given in (3). These boundary conditions account for the production of daughter cells

with age $a = 0$ from all dividing mother cells with age $a > 0$ through the integration over the age variable $a$.

The structure variable $a$ in (2) corresponds to chronological cellular age which we use as a cipher for a number of epigenetic factors. It is possible to include specific epigenetic factors by including a non-constant ageing velocity as in other structured models of physiological processes [49, 50]. However, it is difficult to determine how these epigenetic factors accumulate throughout a cell's lifespan and during treatment. Consequently, including a non-constant ageing velocity would severely complicate the formulation, parametrization, and analysis of (2) and would limit the utility of our simple model, so we only consider chronological age here. While the distinction between disappearance of mother cells and the appearance of daughter cells is natural in age-structured populations such as (2) [50, 51], it results in a strictly negative RHS of (2). Since general ordinary differential equation (ODE) models consider a homogeneous population of cells without accounting for cellular age, there is no distinction made between the division (and subsequent removal) of a mother cell and the appearance of daughter cells. Accordingly, ODE models only include the *net* population gain due to reproduction, i.e. each mother cell producing two daughter cells, which typically results in a non-negative term and differs from the distinction made between division of a mother cell and production of daughter cells in the age structured model (2). This fundamental difference between ODE and structured PDE models results from the inclusion of biological information, such as cellular age, and allows for explicit population heterogeneity in structured PDE models which is generally not possible in the ODE framework.

We assumed that the probability of changing phenotypes depended on the age of the parent cell (i.e., older mother cells are more likely to have daughter cells that switch phenotypes [52, 53], where we recall that we are using chronological age as a surrogate for the degradation of cellular signalling pathways [36, 54–56]). The probability that a cell of age $a$ and phenotype $i$ will create a cell of phenotype $j$ during reproduction is given by $\beta_{ij}(a)$ which leads to the following boundary condition for Eq (2)

$$
\left.\begin{aligned}
A(t,0) &= 2\int_0^\infty [R_A(\bar{A}(t),\bar{B}(t))\beta_{AA}(a)A(t,a) + R_B(\bar{A}(t),\bar{B}(t))\beta_{BA}(a)B(t,a)]\,\mathrm{d}a \\
B(t,0) &= 2\int_0^\infty [R_A(\bar{A}(t),\bar{B}(t))\beta_{AB}(a)A(t,a) + R_B(\bar{A}(t),\bar{B}(t))\beta_{BB}(a)B(t,a)]\,\mathrm{d}a.
\end{aligned}\right\} \quad (3)
$$

The probability of a cell with phenotype $A$ and age $a$ producing two daughter cells with the same phenotype is assumed to be

$$
\beta_{AA}(a) = P_{AA}^* + (P_{AA}^{max} - P_{AA}^*)\exp(-\sigma_A a),
$$

while the probability of a cell of phenotype $B$ with age $a$ producing two cells of phenotype $B$ is assumed to be given by

$$
\beta_{BB}(a) = P_{BB}^* + (P_{BB}^{max} - P_{BB}^*)\exp(-\sigma_B a).
$$

In both cases, $P_{ii}^{max}$ and $P_{ii}^*$ are the maximal and minimal probabilities that a cell with phenotype $i$ produces a cell with the same phenotype, respectively, but are specific to each phenotype. The parameter $\sigma_i$ represents the decay rate of intracellular signalling factors and modulates how ageing impacts the probability of daughter cells retaining the mother cells phenotype. We enforce $\sigma_i > 0$. See Fig A in S1 Text for representative forms and a more in-depth discussion of

$\beta_{AA}$ and $\beta_{BB}$. Nascent cells were restricted to either phenotype $A$ or $B$, i.e.

$$\beta_{AB}(a) = 1 - \beta_{AA}(a) \quad \text{and} \quad \beta_{BA}(a) = 1 - \beta_{BB}(a).$$

We note that the sum

$$A(t,0) + B(t,0) = 2\int_0^\infty [R_A(\bar{A}(t), \bar{B}(t))A(t,a) + R_B(\bar{A}(t), \bar{B}(t))B(t,a)]\mathrm{d}a$$

is the total number of cells being born at time $t$. We give a cartoon schematic of the mathematical model (2) in Fig 1.

We defined a phenotypic *switching strategy* as a pair $(P_{BB}^*, P_{BB}^{max})$ representing the minimal and maximal probability that a daughter cell retains the drug-tolerant phenotype of the parent cell. In what follows, we considered two contrasting and illustrative switching strategies: 1) the *switching strategy* where resistant cells had a high probability of retaining their phenotype if they reproduced early in life (where this probability decreased to 0 as cells age) so $[(P_{BB}^*, P_{BB}^{max}) = (0, 0.9)]$, and 2) the *staying strategy*, where resistant cells were assumed to be unlikely to change phenotype regardless of reproductive age (i.e. $[(P_{BB}^*, P_{BB}^{max}) = (0.95, 1)]$). These strategies, along with a *symmetric* strategy where $(P_{BB}^*, P_{BB}^{max}) = (P_{AA}^*, P_{AA}^{max})$, are illustrated

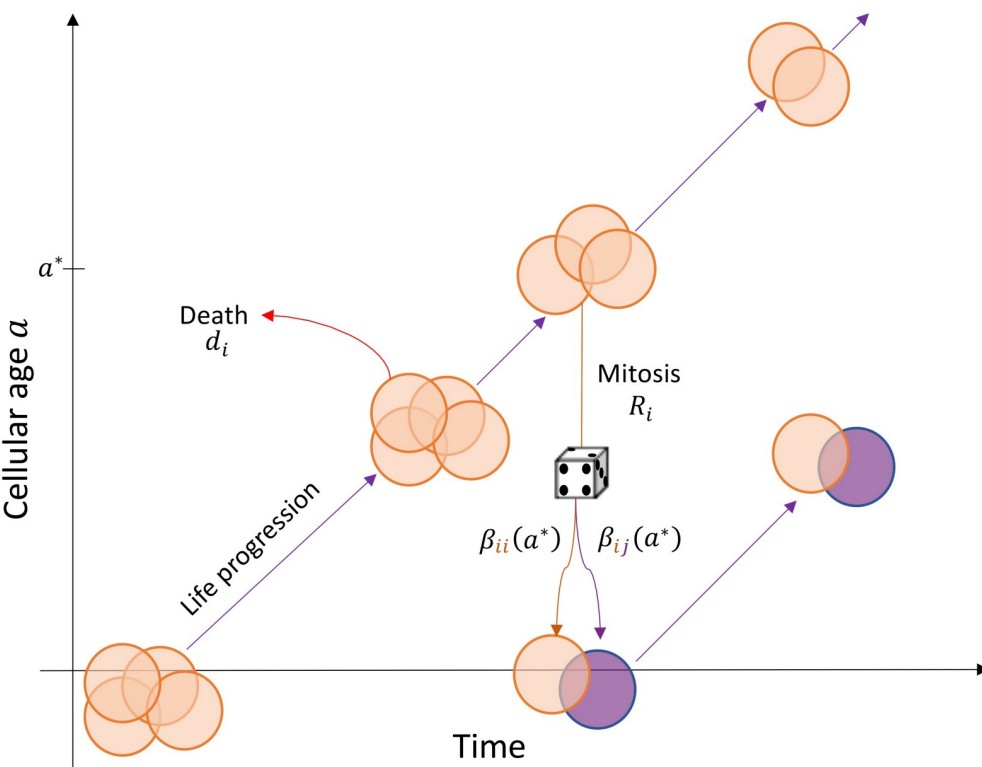

**Fig 1. Schematic of the age structured PDE model (2).** The life progression of a cohort of phenotype $i$ cells with chronological time and age on the $x$ and $y$ axis, respectively. These cells are born with age $a = 0$ and progress through time-age space along the solid lines. The solid lines representing life progression are the characteristic curves of (2). Cells leave the cohort due to death, which occurs at a constant, but phenotype specific, rate $d_i$, or when reproducing at a phenotype specific rate $R_i$. In the figure, a parental cell with age $a^*$ leaves the cohort and reproduces to produce two daughter cells with age $a = 0$. These daughter cells either inherit phenotype $i$ with probability $\beta_{ii}(a^*)$ or change phenotype with probability $\beta_{ij}(a^*)$. In this model formulation, the birth of two daughter cells is modelled as the boundary condition in (3) and corresponds to the appearance of two new cells with age $a = 0$.

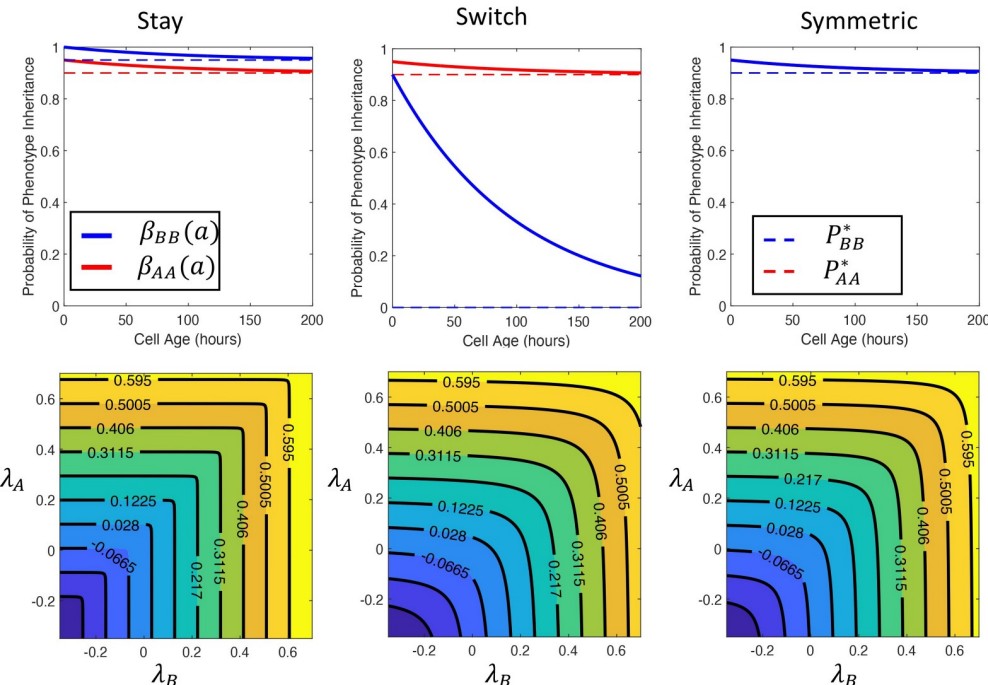

**Fig 2. Switching strategies and the Malthusian parameter $\lambda_P$ as a function of the intrinsic growth rates of sensitive and tolerant cells.** The three switching strategies for the drug-tolerant population are the *stay* strategy $(P^*_{BB}, P^{max}_{BB}) = (0.95, 1)$, the *switching* strategy $(P^*_{BB}, P^{max}_{BB}) = (0, 0.9)$, and the *symmetric* strategy $(P^*_{BB}, P^{max}_{BB}) = (0.9, 0.95)$. In all cases, we fixed $(P^*_{AA}, P^{max}_{AA}) = (0.9, 0.95)$ and $\sigma_A = \sigma_B = 1 \times 10^{-2}$. The probabilities $\beta_{AA}(a)$ and $\beta_{BB}(a)$ for each strategy are shown in the top row. We calculated the Malthusian parameter $\lambda_P$ as detailed in the section *Nonlinear eigenproblem for the Malthusian parameter* in S1 Text and plot the intrinsic growth rate of the mixed population as a function of the intrinsic growth rates of the constituent populations in the second row. The growth rate of the population is increasing along the main diagonal in all cases and the isoclines plot curves of equal population level fitness. In the *stay* strategy, the isoclines are shorter in $\lambda_B$ than in $\lambda_A$ which indicates that $\lambda_P$ is more sensitive to changes in fitness of the tolerant population, $\lambda_B$. Conversely, in the *switch* strategy, the isoclines indicate that $\lambda_P$ is more sensitive to $\lambda_A$. Finally, the Malthusian parameter is symmetric along the main diagonal in the *symmetric* strategy which indicates that fitness of the tolerant or sensitive phenotype equally impacts mixed population fitness.

in the top row of Fig 2. In all cases, we fixed the probability of a drug sensitive cell switching to the drug-tolerant phenotype as $(P^*_{AA}, P^{max}_{AA}) = (0.9, 0.95)$.

Finally, we set

$$A(0, a) = g_A(a) \geqslant 0 \text{ and } B(0, a) = g_B(a) \geqslant 0, \text{ with } \bar{A}(0) < \infty \text{ and } \bar{B}(0) < \infty,$$

to be a biologically-relevant initial condition for the age distribution of cells. We discuss the technical conditions and give expressions for $g_A(a)$ and $g_B(a)$ in section *Initial conditions of the ODE model* in S1 Text.

**Generic model of chemotherapy.** We used a single compartment pharmacokinetic model to include the effects of generic cytotoxic chemotherapy in our mathematical model. We denote the concentration of the chemotherapeutic at time $t$ by $C(t)$. We assumed that the chemotherapeutic was administered intraveneously and has a half life of $t_{1/2}$. This half life determines the linear clearance rate $k_{elim} = \log(2)/t_{1/2}$, so the dynamics of $C(t)$ are given by

$$\frac{\mathrm{d}}{\mathrm{d}t}C(t) = I(t) - k_{elim}C(t) \tag{4}$$

where $I(t)$ models the I.V. administration of the cytotoxic drug during an injection time of $T_{admin}$ administered at times $\{t_i\}_{i=1}^{n}$ and is given by

$$I(t) = \begin{cases} \frac{Dose}{Vol \times T_{admin}} & \text{if} \quad t \in (t_i, t_i + T_{admin}) \\ \\ 0 & \text{otherwise.} \end{cases}$$

We model the effect of chemotherapy through the increase in the death rate of drug-sensitive cells through

$$d_A(t) = d_A + \left(d_A^{max} - d_A\right) \frac{C(t)}{C(t) + C_{1/2}},$$

where the half effect drug concentration is given by $C_{1/2}$ and the maximal death rate of drug-sensitive cells is $d_A^{max}$. We note that it is the ratio of the drug concentration $C(t)$ and the half effect $C_{1/2}$ that completely determine the pharmacodynamics of the therapy in question in our model. While using this simple pharmacodynamic model limits the direct applicability of our work, it allows for the identification of crucial aspects in determining therapeutic effects.

### Effects of phenotypic switching on population fitness

We studied the role of phenotypic heterogeneity on population fitness in the presence of unlimited resources by considering two distinct measures of population fitness: the intrinsic growth rate of the population or Malthusian parameter $\lambda_P$, and the expected number of off-spring or basic reproduction number $R_0$. In structured population models, these quantities are often related through the sign relationship: $\text{sign}(\lambda_P) = \text{sign}(R_0 - 1)$ [57–59]. Precise mathematical formulations and results pertaining to these two metrics are described in section *Model analysis* in S1 Text. In particular, we establish the previously mentioned sign relationship between $\lambda_P$ and $R_0$ in Theorem I in S1 Text. Thus, we can use either $\lambda_P < 0$ or $R_0 < 1$ as thresholds for population growth when later designing a treatment schedule.

A population comprised entirely of drug-sensitive cells has an intrinsic growth rate given by $\lambda_A = r_A - d_A$ (and similarly, $\lambda_B = r_B - d_B$, for a population of entirely drug-tolerant cells, see Proposition B in S1 Text). The cost of resistance was assumed to reduce the intrinsic growth rate in the tolerant population, i.e., $\lambda_B \leqslant \lambda_A$. In a heterogeneous population of cells where cells cannot switch phenotypes, the Malthusian parameter is simply the maximum of growth rates of the constituent populations ($\lambda_P = \max[\lambda_A, \lambda_B]$). However, if cells exhibit phenotype plasticity, then the presence of a less-fit phenotype decreases the fitness of the combined population, and the intrinsic growth rate of the heterogeneous population falls between the growth rate of the constituent populations i.e., $\lambda_P \in (\lambda_B, \lambda_A)$ (Fig 2) for the *switch* and *stay* strategies mentioned earlier as well as the *symmetric* strategy where $(P_{BB}^*, P_{BB}^{max}) = (P_{AA}^*, P_{AA}^{max}) = (0.9, 0.95)$. We first note that the population level fitness is an increasing function of the fitness of the sub-populations, $\lambda_A$ and $\lambda_B$, as we would expect. Further inspection of Fig 2 illustrates how different switching strategies influence the role of fitness increases in each constituent sub-population on fitness of the entire population. In particular, we note that in the *stay* strategy, fitness increases in the drug-tolerant population are more impactful on the entire population than the drug sensitive population, while the opposite is in true for the *stay* strategy.

### Tumour composition evolves during population growth

In nutrient-rich environments, similar to serial replating in *in vitro* experiments, cooperation amongst drug-tolerant cells allows for tumour growth at a rate faster than purely Malthusian growth (Fig B in S1 Text). In the presence of unlimited resources, increasing the death rate $d_A$

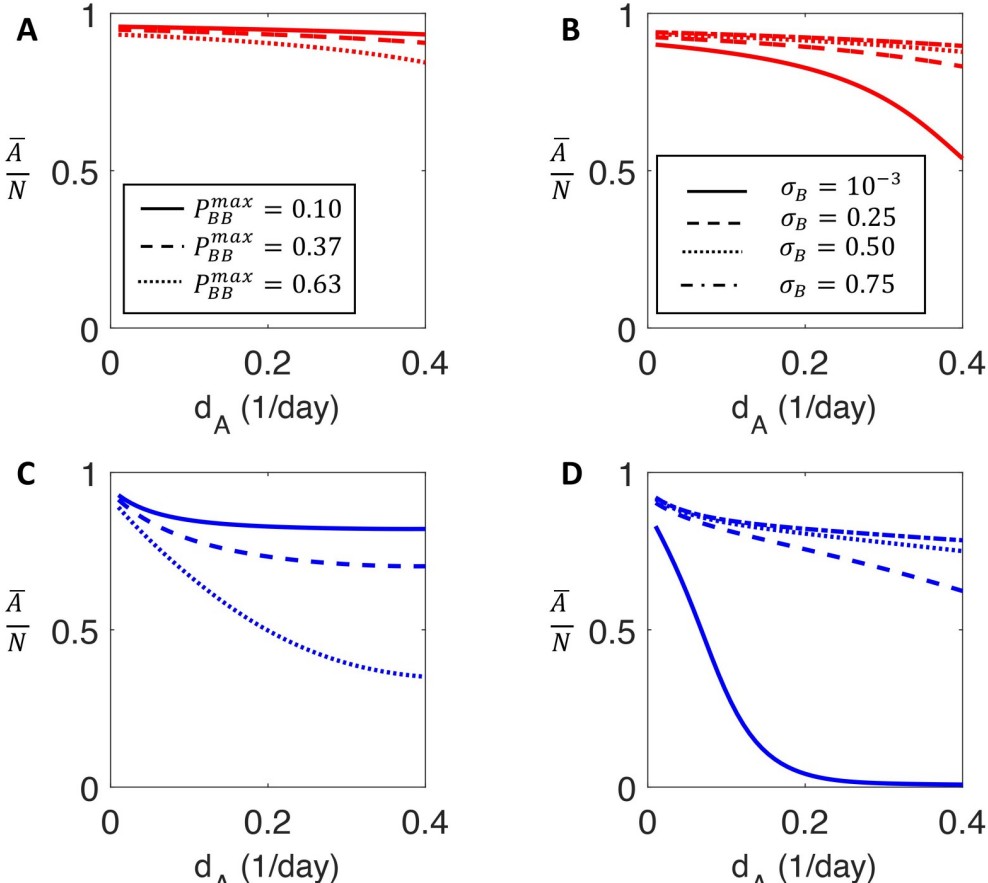

**Fig 3. The proportion of drug-sensitive cells for increasing values of the sensitive cell death rate. A** and **B**) the proportion of the total population that is drug sensitive during Malthusian growth calculated using the stable age distribution detailed in section *Stable age distribution and population proportion* in S1 Text as a function of increasing sensitive death rate $d_A$. **A**) plots the proportion of drug sensitive cells for three values of $P_{BB}^{max} = 0.1, 0.37, 0.63$. **B**) presents the proportion of drug sensitive cells for $\sigma_B = [1 \times 10^{-3}, 0.25, 0.5, 0.75]$. If drug-tolerant cells are unlikely to retain tolerant phenotype upon reproduction, then phenotype switching can partially mitigate the effect of increasing death rate of drug sensitive cells. The proportion of drug-tolerant cells in **A** and **B** compare favourably with the population composition of drug-tolerant cells reported by Sharma et al. [15]. **C** and **D**) the proportion of drug sensitive cells in the limited-resource setting is obtained by simulating the model (2) for 500 days and computing of drug sensitive cells at day 500. **C**) the model predictions for $P_{BB}^{max} = 0.1, 0.37, 0.63$. **D**) the model predictions for $\sigma_B = [1 \times 10^{-3}, 0.25, 0.5, 0.75]$. Phenotypic switching is less capable of mitigating increased death in the resource-limited setting.

of drug-sensitive cells while holding $r_A$ constant acts to decrease the fitness of the drug-sensitive cells (Fig 3A and 3B), corresponding to both a decrease in the relative fitness difference between drug-sensitive and drug-tolerant cells and a decrease in the total population fitness. This is independent of the parameters that determine phenotypic inheritance (see section *Non-linear eigenproblem for the Malthusian parameter* in S1 Text for details). We found that the proportion of sensitive type $A$ cells was consistently higher during unlimited growth than in the resource limited case. This prediction is in line with experimental results where there was a smaller proportion of persister cells during Malthusian growth [20, 39]. Further, the strictly decreasing behaviour of the ratio $\bar{A}/(\bar{A} + \bar{B})$ as $d_A$ increases, so $\lambda_A$ decreases while $\lambda_B$ remains constant, indicates that the fitness difference between phenotypes plays a critical role in determining population composition during Malthusian growth.

In the limited resource situation, contrasting to the Malthusian case, the ratio $\bar{A}/(\bar{A}+\bar{B})$ initially decreases before reaching a plateau and remaining relatively constant as $d_A$ is increased. Accordingly, the relative fitness difference between phenotypes is less important than the probability of phenotypic switching in determining population composition (Fig 3C and 3D). In fact, if the maximal probability of drug-tolerant cells retaining their phenotype ($P_{BB}^{max}$) is sufficiently small, increasing $d_A$ *increases* the proportion of type *A* cells (see section *Phenotype switching may mitigate fitness differences* in S1 Text). Conversely, if drug-tolerant cells are likely to produce drug-tolerant cells via a high probability of phenotypic inheritance, illustrated by the $P_{BB}^{max} = 0.63$ and $\sigma_B = 1 \times 10^{-3}$ cases, the population evolves towards being predominantly drug-tolerant (phenotype B), despite the fact that $\lambda_A > \lambda_B$. Contrary to the unlimited resource case, where the relative fitness between phenotypes is the determining factor, the approximately constant proportion of drug-sensitive cells in the resource-limited setting suggests the importance of resource constraints in driving the establishment of a drug-tolerant population.

**Periodic treatment leads to dominant phenotype switches.**   We next sought to quantify the permanence of treatment resistance as a function of the *switch* and *stay* phenotypic switching strategies discussed earlier. To measure the effectiveness of a given treatment strategy *S* for treatment from $t = 0$ to $t = T_{end}$, we calculated the average total number of cells as a fraction of the population carrying capacity, *K*,

$$\text{Burden}(S) = \frac{1}{T_{end}} \int_0^{T_{end}} \frac{N(\tau)}{K} \, \mathrm{d}\tau = \frac{1}{T_{end}} \int_0^{T_{end}} \frac{\bar{A}(\tau) + \bar{B}(\tau)}{K} \, \mathrm{d}\tau, \tag{5}$$

assuming here the physiologically-realistic finite resource case with phenotypic cooperation. Since we are aiming to avoid competitive release of the resistant sub-population and are primarily interested in a sustainable reduction in tumour size, we consider the cumulative tumour burden over the entire treatment interval. In particular, (5) is related to the objective response rate, or the tumour reduction following therapy, as schedules with a lower tumour burden as defined in (5) would presumably also have a higher objective response rate. Further, we used data from *in vitro* growth assays to parametrize our mathematical model [60]. In these *in vitro* experiments, the population size can be easily measured and used as a proxy for treatment efficacy. However, if we were fitting the model to clinical data rather than *in vitro* data, it would be possible to use more clinically relevant measurements of treatment effect, such as time to disease progression or treatment failure due to resistance. In our framework, treatment resistance was defined as a significantly decreased therapeutic effect on the total population. The robustness of our results when considering different switching strategies is shown in section *General results are robust to parameter variation* in S1 Text.

We simulated a 21-day cyclic chemotherapy with the half-life of the anti-cancer agent set to $t_{1/2} = 6$ hrs, similar to cyclophosphamide, etoposide, and teniposide [61]. For both the *switch* and *stay* strategies, the tumour population eventually developed resistance as the drug-tolerant phenotype became dominant during treatment (Fig 4). We observed that the proportion of drug-sensitive cells in the *switch* population remained above 40% of the total population—at least during the simulated treatment regimen—while the drug-sensitive cells in the *stay* population were effectively driven extinct during treatment. Thus different switching strategies act to either maintain or destroy a treatment-susceptible population. However, in the long term, the *switch* population eventually reverted back to a predominantly drug-sensitive population after treatment was discontinued. Clinically, this corresponds to transient resistance and an eventually re-sensitised population that has been observed in some cancers [8, 15, 16]. This re-sensitization suggests that treatment holidays, where therapy is re-applied after a break may be

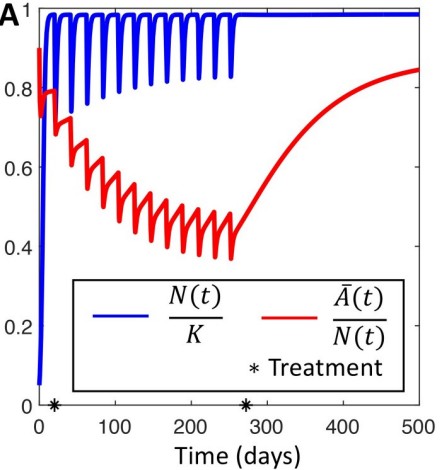
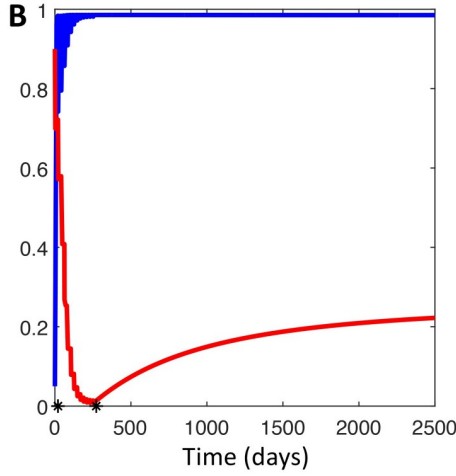

**Fig 4. The effect of switching strategy on treatment efficacy.** The effect of periodic treatment on a population using either the *switch* (**A**) or the *stay* strategy (**B**). Periodic therapy is applied as described in *Periodic Treatment Leads to Dominant Phenotype Switches*. Therapy is applied between days 50 and 275, marked by the black stars. Red curves show the proportion of sensitive cells $\bar{A}(t)/N(t)$. Blue curves shows the dynamics of the total population as a fraction of the population carrying capacity $N(t)/K$. Model parameters used in these simulations are given in Table A in S1 Text.

beneficial, as the *switch* population will eventually return to a mostly drug-sensitive state. Conversely, the *stay* population evolved into a drug-tolerant phenotype dominated state and acquired essentially permanent resistance to therapy. In this case, reapplying the same therapy would be unsuccessful, even after a treatment holiday. These contrasting results demonstrate that different strategies of phenotypic switching can account for two drastically different types of therapeutic resistance by inducing either transient or permanent changes to the population, further underlining the difficulty of designing effective therapies to prevent phenotypic switching. In section *Parameter identifiability during cancer therapy* in S1 Text, we show that the response to treatment can help identify the switching strategy.

## Avoiding resistance to therapy in NSCLC

Lung cancer is the leading cause of cancer-related death in the United States, and non-small cell lung cancer (NSCLC) accounts for 20% of all cancer-related deaths [62]. Nearly two-thirds of NSCLC patients present with surgically unresectable disease and rely on systemic therapies for survival. Better characterisation of NSCLC at the molecular level has resulted in the introduction of a number of targeted therapies that are safer and more effective than standard chemotherapy. However, successful long-term treatment of NSCLC remains hampered by drug resistance [63]. We have recently shown that phenotypic interactions in co-cultured NSCLC spheroids and heterogeneity within patient samples drive a Cooperative Adaptation to Therapy (CAT) [7]. During CAT, cancer cells behave co-operatively to induce drug tolerance in neighbouring cells and thus induce population level treatment resistance. We sought to further quantify the evolution of phenotypic switching in NSCLC using our switching model to better understand treatment failure due to drug tolerance by fitting the model to the Wild Type (WT), Mutation 1 (M1), and Mutation 2 (M2) data from [7]. In the subsequent analysis of our model during therapy, we derived a number of analytical results and expressions that characterise our model-informed treatment schedule. These quantities and their biological

**Table 1. Summary of analytical expressions used to determine model-informed therapy.**

| Parameter | Value | Biological interpretation | Reference |
|---|---|---|---|
| $\lambda_A$ | $r_A - d_A$ | Intrinsic growth rate of drug-sensitive cells | Prop. B |
| $\lambda_B$ | $r_B - d_B$ | Intrinsic growth rate of drug-tolerant cells | Prop. B |
| $\lambda_A^*$ | $r_A - d_A^{max}$ | Treated growth rate of drug-sensitive cells | Definition |
| $\lambda_B(\theta)$ | $r_B f_n(\theta) - d_B$ | Growth rate of drug-tolerant cells with Allee effect | Eq (8) |
| $\varepsilon$ | $\varepsilon \in \left(\frac{r_B}{r_A}, 1\right)$ | Permissible fitness from Allee effect | Chosen |
| $\vartheta_\varepsilon^*$ | $\left[\frac{\varepsilon r_A - r_B}{r_A(1-\varepsilon)}\right]^{1/n}$ | Ratio $\bar{B}/\bar{A}$ to ensure $r_B < \varepsilon r_A$ with Allee effect | Eq (6) |
| $Dose^*$ | $\alpha_T \frac{\lambda_A}{-\lambda_A^*}$ | Minimum dose size to ensure $R_0^* < 1$ | Eq (7) |
| $\theta^*$ | $\left[\frac{-\lambda_B}{\lambda_A}\right]^{1/n}$ | Threshold $\theta^*$ such that $\lambda_B(\theta) < 0$ for $\theta < \theta^*$ | Eq (9) |

interpretation are summarized in Table 1 with the full analytical details given in section *Application to non-small cell lung cancer* in S1 Text.

Beyond the intrinsic heterogeneity within tumours, external factors, including maximally tolerated dosing schedules, can lead to the establishment of a resistant phenotype and limit the effectiveness of therapy. This is clearly clinically disadvantageous as shown in the preceding section. If there were no cooperation amongst drug-tolerant cells, then once the selection pressure of therapy was removed, the population would become re-sensitised to therapy as the fitter, drug-sensitive, phenotype became dominant. Thus, a possible therapeutic strategy is to limit the fitness gain of drug-tolerant cells due to cooperation. In our model formulation, limiting fitness gain due to cooperation is equivalent to limiting cooperation driven increases in reproduction rates modelled by $r_B f_n(\bar{A}(t), \bar{B}(t))$, that is, enforcing $r_B f_n(\bar{A}(t), \bar{B}(t)) < \varepsilon r_A$, where $\varepsilon \in (r_B/r_A, 1)$ measures the allowable amount of fitness gain due to co-operation of drug-tolerant cells. To accomplish this, the ratio of drug-tolerant to drug-sensitive cells, $\theta(t) = \bar{B}(t)/\bar{A}(t)$, must not exceed the threshold ratio $\vartheta_\varepsilon^*$ given by

$$\vartheta_\varepsilon^* = \left[\frac{\varepsilon r_A - r_B}{r_A(1-\varepsilon)}\right]^{1/n}, \tag{6}$$

where we recall that the parameter $n$ determines the strength of the Allee effect in $f_n(\bar{A}(t), \bar{B}(t))$.

Using $\vartheta_\varepsilon^*$, it is possible to schedule therapy to avoid competitive release ("fall then rebound") as drug-sensitive phenotypes switch to drug-tolerant ones and thus maintain a drug-sensitive population (see section *Generic strategy to avoid treatment failure* in S1 Text and [22, 23]).

Here, we detail a strategy for our NSCLC data that simultaneously ensures that the total tumour population decays and the population of drug-tolerant cells remains dependent on the drug-sensitive cells for survival. This strategy requires a delicate balance of maintaining chemotherapeutic concentrations at a large enough value to inhibit growth of the drug-sensitive cells while maintaining the frequency of drug-tolerant cells below a level that induces significant cooperation and the resulting competitive release. In the analytical work that underlies the model-informed schedule, we assumed that $r_B < d_B$. This assumption is satisfied by parameter fitting to the NSCLC data described in section *Application to non-small cell lung cancer data* in S1 Text. In the analytical work, we also assumed that there were ample resources available to use the PDE (2) corresponding to Malthusian growth, although we include the carrying capacity in our simulations. Lastly, we assumed that the chemotherapeutic infiltrated the tumour uniformly and that therapy is administered over a fixed period $T$. We calculate the

chemotherapeutic concentration during metronomic therapy in section *Treatment induced periodic environment* in S1 Text along with the precise analytical results underpinning our strategy.

From a classical population dynamics perspective, if the treated basic reproduction number $R_0^*$ is less than 1, then the tumour population is expected to decay during treatment. In an approximately periodic environment, where chemotherapy is administered every $T$ days, the threshold minimum dose size to ensure that $R_0^* < 1$ is

$$Dose^* = \frac{\lambda_A}{-\lambda_A^*}\alpha_T, \tag{7}$$

where $\lambda_A^* = r_A - d_A^{max} < 0$ is the decay rate of the drug-sensitive population during treatment and $\alpha_T$ is a constant depending on the period of administration $T$. We give a derivation of (7) and the explicit expression for $\alpha_T$ in the section $R_0^*$ *in the treated environment* in S1 Text. To render this threshold clinically relevant, we rewrite Eq (7) as

$$\frac{\lambda_A}{-\lambda_A^*} = \frac{Dose^*}{\alpha_T}.$$

The left hand side above is comprised of patient specific parameters, namely the intrinsic growth rate of the drug susceptible population $\lambda_A$, and the decay rate of the sensitive population during treatment, $\lambda_A^*$. To estimate these quantities, consider two time series, $\bar{A}_i$ and $\bar{A}_i^*$, representing a drug-sensitive population grown in normal media or in the presence of a chemotherapeutic, respectively. Due to phenotypic switching, it is unlikely that these populations are comprised of solely drug susceptible cells, which complicates the estimation of $\lambda_A$ and $\lambda_A^*$ directly from experimental data. Nevertheless, to first approximation, the slope of $\log(\bar{A}_i)$ during the early exponential stage of growth offers an estimate for the intrinsic growth rate $\lambda_A = r_A - d_A$. Assuming, for simplicity, that the chemotherapeutic agent only acts to increase the death rate of cells, then the the slope of $\log(\bar{A}_i^*)$ during exponential decay rate gives an estimate for $\lambda_A^* = r_A - d_A^{max}$. The right hand side of Eq (7) is comprised of parameters describing the properties of the drug as well as the size and frequency of drug administration that can be directly translated to the clinic.

Recalling that the ratio of drug-tolerant to drug-sensitive cells is denoted $\theta(t) = \bar{B}(t)/\bar{A}(t)$, the expression for the fitness of the drug-tolerant population including the Allee effect is

$$\lambda_B(\theta) = r_B f_n(\bar{A}(t), \bar{B}(t)) - d_B. \tag{8}$$

where $f_n(\bar{A}(t), \bar{B}(t))$ models cooperation mediated fitness increases. In particular, we recall that $f_n(\bar{A}(t), \bar{B}(t))$ is a Hill type function with Hill coefficient $n \geqslant 1$. In this context, $n$ can be understood as representing the necessary amount of cooperation between drug-tolerant cells to induce a fitness increase. To inhibit competitive release of drug-tolerant cells (i.e. the observed "fall and rebound" in the NSCLC spheroid data during therapy), we updated our approach to avoid the establishment of the drug-tolerant phenotype by enforcing that $\lambda_B(\theta^*) < 0$ even when considering cooperation. This yields the threshold ratio

$$\theta^* = \left[\frac{-\lambda_B}{\lambda_A}\right]^{1/n}. \tag{9}$$

As $\lambda_B < 0$, the right hand side of Eq (9) is the ratio of the decay rate of drug-tolerant cells to the growth rate of drug-sensitive cells. If the population of drug-tolerant cells decays at a faster rate than the population of drug-sensitive cells grows ($|\lambda_B| > \lambda_A$), then drug-tolerant cells must

outnumber drug-sensitive cells before cooperation will allow for expansion of the drug-tolerant population. In this case, cooperation acts to attenuate the factor by which drug-tolerant cells must outnumber drug-sensitive cells before the drug-tolerant population is self-sustaining, since

$$\left[ \frac{-\lambda_B}{\lambda_A} \right]^{1/n} \leqslant \frac{-\lambda_B}{\lambda_A}.$$

Conversely, if $|\lambda_B| < \lambda_A$, then increasing levels of cooperation necessitates a larger proportion of drug-tolerant cells to permit self-renewal of the treatment resistant population. Once again, the ratio of untreated intrinsic growth rates can be directly estimated from *in vitro* data. In summary, ensuring that $\theta < \theta^*$ is sufficient to avoid the establishment of a resistant population.

## Model-informed treatment drives tumour extinction

Lastly, we combined the strategies ensuring tumour decay or avoiding the establishment of resistance described above to drive long-term treatment effectiveness to docetaxel (see sections *Model informed therapy of other therapeutics* and *Parameter fitting* in S1 Text for similar results for the chemotherapeutics afatanib and bortezomib and details of the parametrization of the pharmacokinetic models for each therapeutic).

In most treatment schedules, docetaxel is administered either weekly or once-every-three-weeks [64], however, it is not obvious that either of these cycle lengths represent optimal treatment periods. Rather, as suggested by Bacevic et al. [23] and others, it may be ideal to dose more frequently and with less intensity to maintain drug pressure on the population. Therefore, we did not *a priori* fix the period of administration *T* to model-informed therapy. Rather, for $T = 1, 2, 3 . . ., 7$ days, we determine the model-informed dose size as

$$\frac{\text{Dose}^*}{C_{1/2} Vol} = \frac{7}{T} \min_{T=1,2,3,...,7} \left[ \left( \frac{\lambda_A}{-\lambda_A^*} \right) (1 - \exp\left[-k_{elim} T\right]) \exp\left(k_{elim} T\right), D_{MTD} \right].$$

where $D_{MTD}$ is the dose size under maximally tolerated dosing. Increasing the density of therapy increases the burden of therapy and may be overwhelmingly toxic. Accordingly, we imposed that the cumulative chemotherapeutic dose under model-informed therapy does not surpass what would be administered in the fixed periodic schedule. For each period *T*, we used the minimal dose size that satisfies (7), as our calculation of $R_0^*$ only identifies a sufficient condition to drive tumour population decay, and any larger dose size potentially allows competitive release of the drug-tolerant population.

We tested the model-informed therapy for each value of $T = 1, 2, 3, . . ., 7$ days and chose the largest therapy period *T* that avoided the establishment of a drug-tolerant population and led to sustained population decay. While the decision to administer therapy on each treatment day *nT* is dependent on the ratio $\theta(t) < \theta^*$, and so the tumour micro-environment may not be precisely periodic, our results indicate that combining the two model-informed constraints successfully drives tumour extinction.

We compared our model-informed therapy to periodic dosing administered every 7 days and found that informed therapy performed comparatively to periodic dosing during the initial stage of therapy (Fig 5). However, the benefit of our model-informed therapy becomes apparent when inspecting the behaviour of the treated tumour over longer periods: the fixed dosing schedule allowed for the establishment of a drug-tolerant phenotype and the eventual loss of effectiveness of therapy, while the model-informed therapy maintained a drug-sensitive population and led to sustained tumour decay. In fact, the stable oscillations at (or below) the

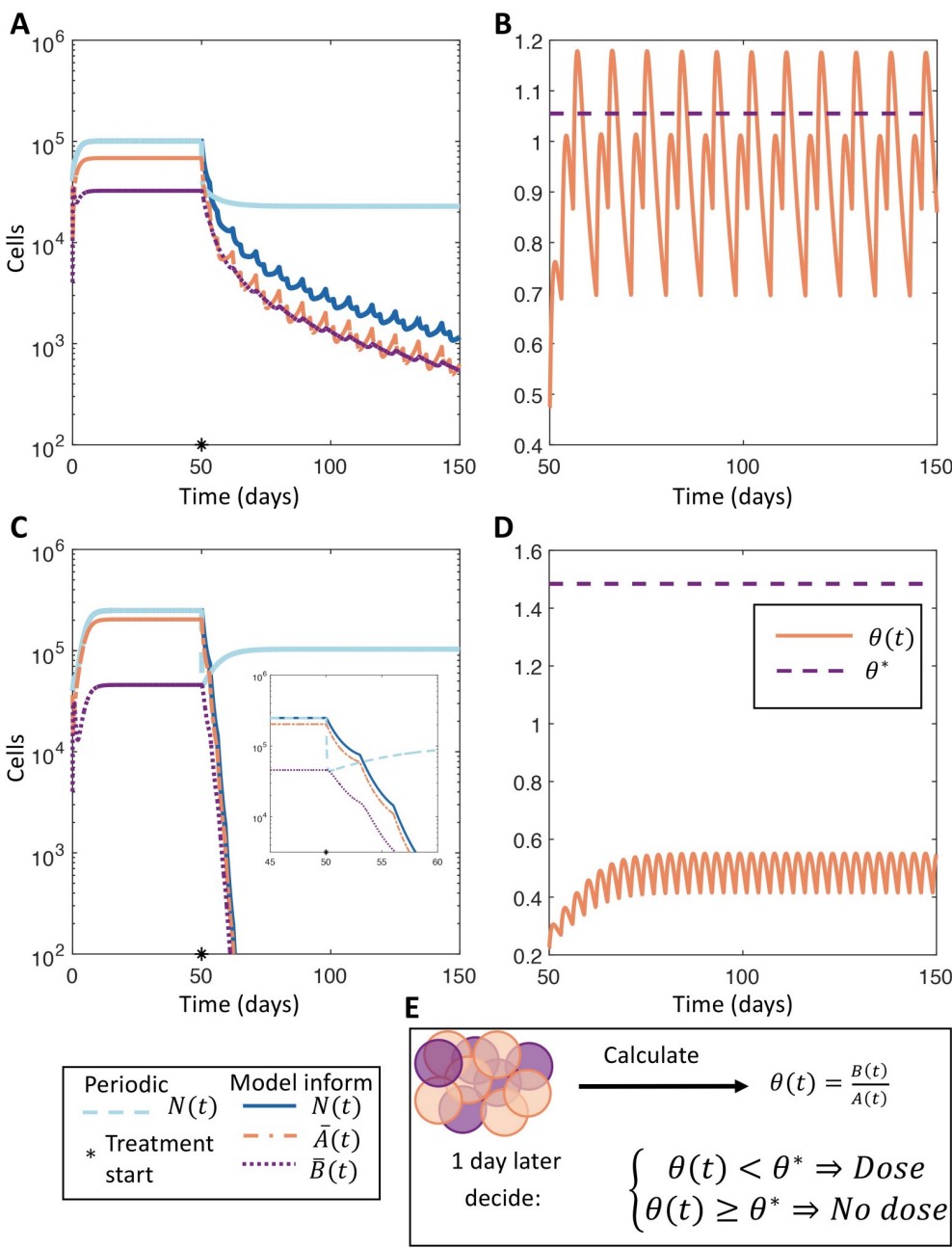

**Fig 5. Comparing model-informed and periodic dosing treatment regimens. A** and **C)** Comparison of model-informed therapeutic strategy with $T = 3$ against periodic treatment with docetaxel for the WT and M1 populations, respectively. The total tumour populations under periodic or model-informed therapies are given in dashed grey or in solid blue, respectively. The drug sensitive $(\bar{A}(t))$ and drug-tolerant $(\bar{B}(t))$ sub-populations under model-informed therapy are shown in dot-dashed orange or dotted purple. The beginning of treatment on day 50 is denoted by a black star. The inset in **C** shows the rapid decay of the M1 tumour population during therapy. **B** and **D)** The ratio $\theta(t) = \bar{B}(t)/\bar{A}(t)$ during model-informed therapy in solid orange and the threshold ratio $\theta^*$ in dashed orange for the WT and M1 populations respectively. **E)** illustrates the model-informed therapy where $\theta(t)$ is used to decide if therapy is given or not. The model parameters used in this simulation are given in Tables C and D in S1 Text.

threshold ratio $\theta^*$ in Fig 5B and 5D combined with the exponential decay of $N(t)$ shown in Fig 5A and 5C illustrate the efficacy of the model-informed therapy to preserve a sufficiently large population of sensitive cells while driving tumour extinction. The model-informed therapy is designed to consistently have a large enough population of sensitive cells, characterised by $\theta(t) < \theta^*$, to avoid resistance while imposing enough treatment pressure to ensure $R_0^* < 1$ during therapy (see sections $R_0^*$ *in the treated environment* and *Limiting cooperation of drug tolerant cells* in S1 Text for details). Consequently, the observed exponential decay is precisely what is predicted from the design of model-informed therapy.

As before, we computed the effectiveness of therapy using Eq (5). The ratio of tumour burden in the model-informed therapy to periodic therapy was 0.5598 and 0.7600 for the WT and M1 cells studied (see Materials and methods), respectively, over 100 days of treatment. The effectiveness of model-informed therapy becomes more pronounced when considering longer treatment intervals, as the model-informed therapy drives sustained tumour decay. This result clearly demonstrates that model-informed therapy significantly outperforms periodic therapy, and is consistent across all populations and therapeutics considered, demonstrating the robust efficacy of this adaptive and model informed approach to maintaining drug-tolerant phenotypes.

## Discussion

Despite the introduction of novel targeted therapies and increased characterisations of individual patient's genetic landscapes, drug resistance continues to drive treatment failure. This suggests that identifying and understanding non-genetic factors contributing to drug therapy tolerance is crucial to providing better care. In this work we proposed a simple quantitative model of stochastic phenotype switching in the context of cancer. Our model is comprised of two non-local age structured PDEs that incorporate phenotypic switching through non-local boundary terms. Specifically, phenotypic switching is described as a random process where the probability of inheriting the parents' phenotype is a decreasing function of cellular age at reproduction. This mapping from age to switching probability generalises the role of molecular switching mechanisms and the inheritence of signalling factors in phenotype determination. In this sense, we have studied the role of phenotypic 'memory' governed by the inheritance of intracellular factors, similar to the biological phenomenon observed by Yang et al. [36] where inheritance of signalling factors such as p53 and mitogen can predispose daughter cells towards quiescence or proliferation. Recent experimental work has implicated these signalling factors and the resulting non-genetic memory in response to anti-cancer treatment. In particular, resistance to targeted therapies in NSCLC has been shown to result from a series of gradual epigenetic and genetic adaptations to treatment induced selection pressures [65].

Those experimental results have specifically identified the role of treatment induced stress on mother cells as a determining factor in daughter cell's adoption of a "persister" like phenotype [52, 53, 66–68]. In previous theoretical work we demonstrated that the precise mechanisms governing phenotype switching determine the rate of extinction under cytotoxic therapy [13]. Thus, molecular switching mechanisms may be subject to evolution by natural selection. The model presented here represents a more general framework to further explore this phenomenon, and to extend it through the introduction of phenotypic memory, by specifying switching dynamics in a functional, rather than network-defined, form. In this sense, our work addresses similar questions to [27, 28] although via a different and complementary axis, particularly in our analytical results and the development of the model-informed treatment.

Given the assumption of unlimited resources, we derived expressions for the Malthusian parameter and basic reproduction number, and established the classic sign relationship

between these two measures of population fitness. From the structured PDE model (2), we derived an equivalent ODE model describing the dynamics of drug-tolerant and sensitive populations to study the impact of resource availability and intra-phenotype cooperation on population growth. This allowed us to show that competition for limited resources facilitates the establishment of a less fit phenotype. Incorporating a phenomenological model of cytotoxic therapy, we showed that the phenotype switching strategy of the drug-tolerant population (to either preferentially inherit or relinquish the parent cells phenotype) determined the type of treatment resistance observed. In particular, our mathematical model can reproduce both transient drug resistance or epigenetic permanent resistance by only changing the switching strategy of the drug-tolerant population. Leveraging this, we proposed a treatment schedule that exploits the population composition to avoid the establishment of treatment resistance.

Importantly, we then applied our model to understand the development of treatment resistance within *ex vivo* NSCLC tumour spheroids to understand the impact of phenotypic switching on response to treatment. When exposed to chemotherapeutics, genetically identical NSCLC populations were found to exhibit a "fall then rebound" behaviour indicative of phenotypic resistance. We derived the basic reproductive number in the context of periodic treatment and determined a therapy schedule that avoided the establishment of resistance and exhibited sustained tumour decay. The NSCLC data and our results underline that phenotypic switching may be occurring in a genetically identical population of NSCLC cells and may be driving treatment resistance. It is thus important to quantify its presence and impact of on treatment scheduling. In this work, we presented a mathematical model to understand phenotypic heterogeneity and derived a model-informed strategy to mitigate –and potentially avoid– phenoytpically driven treatment resistance.

Our phenomenologically-based model is simple. Consequently, our results must be evaluated in light of the many assumptions and limitations of our model, and remain to be further validated in experimental systems. We also made an important assumption that cancer cells are either entirely drug-tolerant or drug-sensitive. While this assumption of a discrete phenotype landscape simplifies the mathematical modelling, it is not biologically realistic. Furthermore, the results of Vander Velde et al. [65] suggest implementing a continuous phenotype landscape in our model as well as extending our analysis to study combination therapies, strategies for drug combination, and the continued evolution of treatment resistance. Moreover, we do not consider the role of spatial and metabolic heterogeneity [69–71], drug infiltration [72, 73], nor the role of other cells in the tumour micro-environment [74].

These limitations notwithstanding, our work identifies the role of stochastic switching in therapeutic resistance, explicitly incorporates non-genetic inheritance, or phenotypic memory, in a physiologically structured mathematical model, and highlights the role of mathematical modelling in understanding and developing evolutionary-inspired therapeutic strategies.

## Materials and methods

### Non-small cell lung cancer data

We used the previously published *in vitro* growth assay data from Craig et al. [7] to parametrize our mathematical model. Briefly, in their work, the parental (WT) cell line was derived from KRas-G12D, p53−/−, Dicer1f/− genotype lung tumours and mutants (M1 and M2) were obtained through transfection to Dicer1+/+ and Dicer1-/- using CRISPR-Cas9 [75]. Cells were plated as tumour spheroids on NanoCulture plates and population growth without and with drug was assessed via flow cytometry on days 1, 3, 5, and 7 [7].

In the untreated experiments, Craig et al. [7] cultured a genotypically homogeneous population of NSCLC cells for 7 days. In the treated experiments, after 72 hours of growth in

untreated medium, the authors bathed the population of cells in a constant and lethal concentration of one of three chemotherapeutics (docetaxel, afatinib, or bortezomib) and counted the number of surviving cells. As the anti-cancer drug concentration is constant, we assume that the observed "fall and rebound" behaviour is not driven by the proliferation of a drug-sensitive population, but rather due to the expansion of a drug-tolerant population, similar to the phenotypic resistance observed in numerous studies [8, 15, 16]. While the drug-tolerant population may have arisen due to genetic mutations, the short treatment time of 96 hours suggests the expansion of a previously established drug-tolerant phenotype. We report results for the WT and Mutation 1 (M1) lineages treated with docetaxel in the main text with similar results for WT, M1 cells treated with afatinib and bortezomib, as well as a separate population Mutation 2 (M2) cells shown in section *Model informed therapy for other therapeutics* in S1 Text.

## Numerical simulation of phenotypic switching model

Eq (2) is a system of coupled non-local PDEs for the cell densities $A(t, a)$ and $B(t, a)$. Rather than implementing these PDEs numerically, we note that we are primarily interested in the number of drug-sensitive and drug-tolerant cells, given by

$$\bar{A}(t) = \int_0^\infty A(t, a) \mathrm{d}a \quad \text{and} \quad \bar{B}(t) = \int_0^\infty B(t, a) \mathrm{d}a.$$

For $L_1$ initial data, the theory of transport equations ensures that these integrals are finite for $t > 0$ [51]. Therefore, rather than solving the system of coupled PDEs and integrating over age to compute $\bar{A}(t)$ and $\bar{B}(t)$, we derive an equivalent finite dimensional system of ordinary differential equations for the populations $\bar{A}(t)$ and $\bar{B}(t)$. The derivation uses Leibniz's integral rule and integration by parts and is detailed in section *Ordinary differential equations for $\bar{A}(t)$ and $\bar{B}(t)$* in S1 Text. Incorporating phenotypic switching through the boundary conditions necessitates two extra ODEs for the proportion of drug-sensitive or drug-tolerant cells retaining their phenotype. The resulting ODE model is

$$\frac{\mathrm{d}}{\mathrm{d}t}\bar{A}(t) = -\left[R_A(\bar{A}(t), \bar{B}(t)) + d_A + (d_A^{max} - d_A)\frac{C(t)}{C(t) + C_{1/2}}\right]\bar{A}(t)$$
$$+ 2R_A(\bar{A}(t), \bar{B}(t))N_{AA}(t) + 2R_B(\bar{A}(t), \bar{B}(t))[\bar{B}(t) - N_{BB}(t)]$$

$$\frac{\mathrm{d}}{\mathrm{d}t}\bar{B}(t) = -[R_B(\bar{A}(t), \bar{B}(t) + d_B]\bar{B}(t) + 2R_A(\bar{A}(t), \bar{B}(t))(\bar{A}(t) - N_{AA}(t))$$
$$+ 2R_B(\bar{A}(t), \bar{B}(t))N_{BB}(t)$$

$$\frac{\mathrm{d}}{\mathrm{d}t}N_{AA}(t) = P_{AA}^{max}\left[2R_A(\bar{A}(t), \bar{B}(t))N_{AA}(t) + 2R_B(\bar{A}(t), \bar{B}(t))(\bar{B}(t) - N_{BB}(t))\right]$$
$$- \left(R_A(\bar{A}(t), \bar{B}(t)) + d_A + (d_A^{max} - d_A)\frac{C(t)}{C(t) + C_{1/2}}\right)N_{AA}(t)$$
$$+ \sigma_A(P_A^*\bar{A}(t) - N_{AA}(t))$$

$$\frac{\mathrm{d}}{\mathrm{d}t}N_{BB}(t) = P_{BB}^{max}[2R_A(\bar{A}(t), \bar{B}(t))(\bar{A}(t) - N_{AA}(t)) + 2R_B(\bar{A}(t), \bar{B}(t))N_{BB}(t)]$$
$$- (R_B(\bar{A}(t), \bar{B}(t)) + d_B)N_{BB}(t) - \sigma_B N_{BB}(t) + \sigma_B P_B^*\bar{B}(t),$$

$$\frac{\mathrm{d}}{\mathrm{d}t}C(t) = I(t) - k_{elim}C(t)$$

where we use the single compartment model for $C(t)$ and use initial conditions corresponding to tumour populations in exponential growth that are given in section *Initial conditions of the ODE model* in S1 Text.

## Model parametrization to NSCLC data

To fit the mathematical model to the NSCLC *in vitro* data, we fix $P^*_{AA} = 0$ and $P^{max}_{AA} = 0.95$, set $\sigma_A = \sigma_B = 1 \times 10^{-2}$ hours$^{-1}$ and $d_A = d_B$, and account for the fitness cost of resistance by enforcing $r_B \leqslant r_A$. The parameters remaining to be fit control either population growth ($r_A$, $r_B$, and $d_A$), or the probability of retaining the drug-tolerant phenotype ($P^*_{BB}$ and $P^{max}_{BB}$). Thus, there are five parameters to be fit to experimental data.

We show in section *Parameter identifiability during cancer therapy* in S1 Text that these parameters may not be identifiable for untreated data. In particular, for a given pair $(P^*_{BB}, P^{max}_{BB})$, it is possible to fit the parameters $r_A$, $r_B$ and $d_A$ can be chosen to fit experimental data equally well in the absence of treatment. However, the role of the parameters $(P^*_{BB}, P^{max}_{BB})$ becomes evident once therapy is administered and the previously indistinguishable curves become distinct. Therefore, we simultaneously fit the untreated and docetaxel data from [7] to determine the five parameters to be fit.

During the treated experiments, the cells are continuously bathed in lethal concentrations of each chemotherapeutic, so we model the death rate of the drug-sensitive cells during fitting as

$$d_A(t) = \begin{cases} d_A & \text{if} \quad t < t_{treat} \\ d_A^{max} & \text{if} \quad t \geqslant t_{treat}. \end{cases}$$

For treated and untreated time series data $\{\text{Data}_i\}_{i=1}^n$, we fit the parameters $r_A, r_A, d_A = d_B, d_A^{max}, P^*_{BB}$ and $P^{max}_{BB}$ by minimizing

$$\text{Error}(r_A, r_B, d_A, d_A^{max}, P^*_{BB}, P^{max}_{BB}) = \sum_{i=1}^n \left( N(t_i) - \text{Data}_i \right)^2 \tag{10}$$

where $N(t_i) = \bar{A}(t_i) + \bar{B}(t_i)$ is the total number of cancer cells predicted by the mathematical model. We used the Matlab [76] algorithm *fmincon* to minimize (10) with 15 initial starting points in parameter space. The results of our fitting to the untreated and docetaxel data are shown in Fig F in S1 Text.

Having fit the parameters $r_A, r_B, d_A = d_B, d_A^{max}, P^*_{BB}$ and $P^{max}_{BB}$ to the untreated and docetaxel treated population data, we fix the tumour growth parameters $r_A, r_B, d_A = d_B$, and only fit $d_A^{max}, P^*_{BB}$ and $P^{max}_{BB}$ for the data from experiments with afatinib and bortezomib. We do not refit the growth rate $r_B$, both to avoid overfitting, and as cancer cells with a drug-tolerant phenotype have exhibited cross-resistance to other chemotherapeutics [16]. We list the tumour growth and switching parameters in Tables B and C in S1 Text.

## Supporting information

**S1 Text. Supporting information file.** Supporting mathematical analysis, tables, and figures.

**Fig A. Phenotypic switching probability and relative fitness gain**. Figure **A** shows a representative form of the function $\beta_{ii}(a)$ that represents the probability that a mother cell with phenotype $i$ and age $a$ bequeaths it's phenotype to the daughter cells. Figure **B** shows the

frequency dependent fitness increase function $f_n(\theta)$ for $n = 1, 2, 10$ used to model fitness increases of the drug tolerant phenotype due to the Allee effect.

**Table A. The generic model parameters**. The generic model parameters used to illustrate the impact of phenotypic switching on treatment resistance.

**Fig B. A comparison of growth rates for different growth functions $f_n$, $n = 1, 2, 3, 10$, against Malthusian growth**. The "no Allee" curves correspond to no frequency dependent fitness increase and $f_n = 1$. Figure **A** shows the population evolution from an initial population comprised of 100 drug sensitive cells and one drug tolerant cell for the generic parameters in Table S1 obtained by simulating (S15). Figure **B** shows the population evolution from an initial population comprised of 1 drug tolerant cell and 100 drug tolerant cells for the generic parameters in Table A obtained by simulating (S15).

**Fig C. The proportion of drug-sensitive cells for increasing values of the sensitive cell death rate**. **A** and **B**) the proportion of drug sensitive cells in the limited-resource setting is obtained by simulating the model (S2) for 500 days and computing of drug sensitive cells at day 500. **A**) the model predictions for $P_{BB}^{max} = 0.05, 0.13, 0.22$. **B**) the model predictions for $\sigma_B = [1 \times 10^{-3}, 0.25, 0.5, 0.75]$ with $P_{BB}^{max} = 0.6$.

**Fig D. Fitting of the mathematical model to the Dingli et al. 2009 [60] data for a variety of fitting strategies**. Figure **A** shows the fitting of the mathematical model (S2) to the Dingli et al. [60] data for the 8 different switching strategies given in the text. Figure **B** shows the regrouping of same 8 strategies after 2 applications of therapy. The parameters used in these simulations are given in Table B.

**Table B. The tumour growth parameters obtained by fitting (S2) to the Dingli et al. [60] data for the 8 different switching strategies**.

**Fig E. The effect of adjustable therapy on a population using either a *switch* or *stay* strategy**. The *switch* population is shown in figure **A**) and the *stay* population is shown in figure **B**). In both cases, treatment is applied between the black stars on days 20 and 272. The red curve shows the proportion of drug sensitive cells $\bar{A}(t)/N(t)$ and the blue curve shows the dynamics of $N(t)$. The parameters used in these simulations are given in Table A.

**Fig F. Fitting results of Equation (S2) to the WT and M1 data from [7] treated with docetaxel**. The model fits to the WT and M1 data are shown in Figures **A** and **B**, respectively. In all cases, the untreated data is given by the black stars while the untreated simulation is in solid blue. The docetaxel treated data is given by the hollow circles and the treated simulation is in dashed blue. The parameters used in these simulations are given in in Tables C and D.

**Fig G. Fitting results of Equation (S2) to the WT, M1 and M2 data from [7] treated with afatinib and bortezomib**. The top row shows the model fits to the WT, M1, and M2 data treated with afatinib, respectively. The bottom row shows the model fits to the WT, M1, and M2 data treated with bortezomib, respectively. The parameters used in these simulations are given in Tables C and D.

**Table C. The switching parameters for WT, M1, and M2 cell lines obtained by fitting (S2) to the tumour growth data**.

**Table D. The tumour growth parameters for the WT, M1, and M2 type cells obtained by fitting (S2) to the tumour growth data**.

**Table E. The effectiveness of model-informed therapy when compared to periodic dosing over 150 days of therapy**.

**Fig H. Comparing model informed therapy and periodic dosing for afatinib** Figures **A** and **C** compare model-informed therapeutic strategy with $T = 3$ against periodic treatment with afatinib for the WT and M1 populations, respectively. The total tumour populations

under periodic or model-informed therapies are given in dashed grey or in solid blue, respectively. The drug sensitive $(\bar{A}(t))$ and drug tolerant $(\bar{B}(t))$ sub-populations under model informed therapy are shown in dot-dashed orange or dotted purple. The beginning of treatment on day 50 is denoted by a black star. The inset figure in Figure **C** shows the rapid decay of the M1 tumour population during therapy. Figures **B** and **D** show the ratio $\theta(t) = \bar{B}(t)/\bar{A}(t)$ during model-informed therapy in solid orange and the threshold ratio $\theta^*$ in dashed orange. Figure **E** illustrates the model-informed therapy where $\theta(t)$ is used to decide if therapy is given or not. The model parameters used in this simulation are given in Tables C and D.

**Fig I. Comparing model informed therapy and periodic dosing for bortezomib**. Figures **A** and **C** compare model-informed therapeutic strategy with $T = 3$ against periodic treatment with bortezomib for the WT and M1 populations, respectively. The total tumour populations under periodic or model-informed therapies are given in dashed grey or in solid blue, respectively. The drug sensitive $(\bar{A}(t))$ and drug tolerant $(\bar{B}(t))$ sub-populations under model informed therapy are shown in dot-dashed orange or dotted purple. The beginning of treatment on day 50 is denoted by a black star. The inset figure in Figure **C** shows the rapid decay of the M1 tumour population during therapy. Figures **B** and **D** show the ratio $\theta(t) = \bar{B}(t)/\bar{A}(t)$ during model-informed therapy in solid orange and the threshold ratio $\theta^*$ in dashed orange. Figure **E** illustrates the model-informed therapy where $\theta(t)$ is used to decide if therapy is given or not. The model parameters used in this simulation are given in Tables C and D.

(PDF)

## Acknowledgments

Portions of this work were completed while TC, DN, and ARAA participated in the thematic semester in Mathematical Biology at the Institut Mittag-Leffler. TC is grateful for many useful conversations with Tony Humphries.

## Author Contributions

**Conceptualization:** Tyler Cassidy, Daniel Nichol, Mark Robertson-Tessi, Morgan Craig, Alexander R. A. Anderson.

**Data curation:** Tyler Cassidy, Morgan Craig.

**Formal analysis:** Tyler Cassidy, Daniel Nichol.

**Funding acquisition:** Morgan Craig, Alexander R. A. Anderson.

**Investigation:** Tyler Cassidy, Daniel Nichol.

**Methodology:** Tyler Cassidy, Daniel Nichol.

**Software:** Tyler Cassidy.

**Supervision:** Morgan Craig, Alexander R. A. Anderson.

**Visualization:** Tyler Cassidy, Morgan Craig, Alexander R. A. Anderson.

**Writing – original draft:** Tyler Cassidy, Daniel Nichol, Mark Robertson-Tessi.

**Writing – review & editing:** Tyler Cassidy, Daniel Nichol, Mark Robertson-Tessi, Morgan Craig, Alexander R. A. Anderson.

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
