## [Decision Letter · Decision Letter 0]

10 May 2021

Dear Dr Cassidy,

Thank you very much for submitting your manuscript "The role of memory in non-genetic inheritance and its impact on cancer treatment resistance" for consideration at PLOS Computational Biology. As with all papers reviewed by the journal, your manuscript was reviewed by members of the editorial board and by several independent reviewers. The reviewers appreciated the attention to an important topic. Based on the reviews, we are likely to accept this manuscript for publication, providing that you modify the manuscript according to the review recommendations.

Sincerely,

Dominik Wodarz

Associate Editor

PLOS Computational Biology

Natalia Komarova

Deputy Editor

PLOS Computational Biology

[LINK]

Reviewer's Responses to Questions

**Comments to the Authors:**

Reviewer #1: The authors have developed a stochastic model that tracks phenotypic switching of cells between their drug-resistant and drug-sensitive states, and demonstrate how this switching can influence the timescale of emergence and maintenance of drug tolerance in a phenotypically heterogeneous population. Based on fitting their simulation profiles to in vitro data, they also identify therapeutic strategies that can lead to sustained decay of tumor size without exhibiting long-term resistance. The study is well-done overall and caters to the emerging theme of non-genetic heterogeneity in enabling drug resistance/tolerance. I have the following request for authors to clarify some of their model assumptions and interpretations:

1. If I understood correctly, the authors allow for phenotypic switching only at cell division, right? They should include a schematic as Fig 1 to explain more clearly their modeling framework. Also, is one of the inherent assumptions that both drug-tolerant and drug-sensitive cells have equal switching rate (or propensity in a continuum framework) to one another? Is age the only parameter that influences the switching rate/propensity? Also, the authors should explain the existence of terms R_A (A(t), B(t)) and R_B (A(t) in equation (3).

2. In their modeling framework, is the role of ‘age’ similar to that of lineage tracing/barcoding a cell, i.e. counting for how many simulation time steps has an individual cell been around? What is the connection of age with permanent vs. temporary resistance in the framework?

3. How are age and ‘memory’ related? Do the authors define ‘memory’ of a cell as its ability to maintain a phenotype upon cell division? Usually, the concept of ‘cell memory’ is invoked upon to indicate hysteresis in a given system (Jolly & Celia-Terrassa, J Clin Med 2019).

4. The authors should clarify how many different parameters were fitted to the experimental data, and how many time points and conditions are needed to identify those number of parameters, without overfitting?

5. The authors claim using ref 44 and 45 that with age, the rate of switching increases. However, both ref 44 and 45 do not seem to show this directly. Also, both of them are in bacterial systems, not cancer cells. Can the authors provide stronger evidence for this key assumption in their model? Also, the authors provide the results for ‘stay’ strategy using values of P_BB and P_BB_max close to one another as well as close to 1; what happens for values say 0.45, 0.5?

6. The authors should comment on similarities and differences of their model formulation and key results with other recent efforts – Sahoo et al. bioRxiv 2021, Gunnarsson et al. J Theor Bio 2020).

Reviewer #2: The authors propose a mathematical model to investigate the role of phenotypic plasticity in treatment resistance, and to investigate treatment strategies to avoid establishment of drug-resistant phenotypes. It is shown that a model-informed therapy could drive tumor to extinction while preventing the risk of development of resistance.

The paper is very well written and the mathematics is elegant and impressive. The model is simple but effective to illustrate important biological mechanisms. I only list very minor details below, that the authors may or may not take into account.

Minor comments.

- Supplementary vs Supplemental throughout the text

- Line 105: could be worth to explain the overline notation here already; another minor comment is that it could be worth to mention how reproduction is intended, i.e., that at rate R_A cells die and two daughter cells are born.

- Line 114: maybe mention that n is a parameter that describes the type of response

- Line 127: leads to THE following

- Eq (3): may be clearer to use brackets to isolate the argument of the integral

- P. 4, before the definition of beta_AA, I would find it clearer to use “is assumed to be” rather than “given by”, to make it clear that this is a model assumption. Also, please mention that sigma are positive parameters (incidentally I was curious why P^* is not denoted by P^{min}, but not necessary to change)

- I find Figure 1 little informative, considering that the shown behavior is quite intuitive. Maybe some extra explanation to stress what you want to show?

- Line 167: that -> than

- Fig 2, caption: isn’t this for increasing values of the SENSITIVE cell death rate? In the legend, I find 11/30 and 19/30 less intuitive to interpret as the decimal notation, I’m not sure if there was a reason for this but it is unclear; it may be worth to use the same scale in vertical axis in panels A-B and panels C-D

- Line 184-186: how is the sigma_B fixed?

- Line 187: …of the POPULATION carrying capacity $K$ (in order to also define K)

- Line 194: I am not familiar with the term “objective response rate”

- Line 207: “the drug-tolerant population became dominant”: this sentence is unclear to me, as the left panel shows that the proportion of drug tolerant cells is only 20%, hence it doesn’t seem to me to be dominant. Maybe clarify what you mean?

- Line 208: in the switch population in Fig 3A, sensitive cells seem to remain above 40% (not just above 20%)

- Figure 3 (and results): it is unclear at this stage what parameters are used for the drud-sensitive population (beta_AA). May be useful to include a table?

- Table 1, seventh entry: Ratio B/A (rather than A/B)?

last entry: “such that lambda_B(theta)<0”, add: for theta < theta^*

- Eq (5): is n the same parameter defining the Allee effect? If so, it may be useful to briefly recall it

- Line 270: a approximately -> an approximately

- Line 384: it’s -> its

- Line 434 “Generic model of chemotherapy”: I think this section would better be located before the “Numerical simulation of phenotypic switching model”, as it defines the variable C(t) and related quantities, which are otherwise not defined in the ODE system presented at p. 15. I would also specify that this describes the standard PERIODIC treatment mentioned in the results.

- Equation after (9): bracket is missing from interval

- Lines 465 and 467: “r_A” is listed twice in the two lists – typo?

In the Supplementary Material:

- After (S5), “As expected” sounds strange as I thought this was the assumption leading to the choice of the beta functions

- P. 1 line 39: “after 1 day will HAVE”

- P. 2 line 76, “nutrient” (typo)

- P. 3 line 77 “carrying”

- Four lines after (S8): relative fitness OF these cells

- Equation after (S10): a closing bracket is missing in the interval

- P. 5, three lines after definition of N_AA : not sure if “either” is in the correct position in the sentence

- Equation after (S14): R_I should be R_A ?

- P. 7 second line of the equation for N_AA in (S15): there is an argument “ts” (typo)

- P. 9 line 166: an stable -> a stable (typo). You should also probably mention that the eigenproblem is studied for the linearization of (S2), or alternatively for constant growth rates rA and rB ?

- P. 19 line 371: extra closing bracket

- P. 24, line after (S32): are you here assuming unconstrained growth R_i = r_i?

- P. 25 last equation: comma rather than full stop

- P. 28, line 4: f_N -> f_n

Reviewer #3: Attachment

**Have the authors made all data and (if applicable) computational code underlying the findings in their manuscript fully available?**

Reviewer #1: Yes

Reviewer #2: None

Reviewer #3: **No: **The data is available but the code used to carry out the modelling has not been linked

PLOS authors have the option to publish the peer review history of their article (what does this mean?). If published, this will include your full peer review and any attached files.

Reviewer #1: No

Reviewer #2: No

Reviewer #3: **Yes: **Stephanie Owen, Jacob G Scott

Figure Files:

Data Requirements:

Reproducibility:

References:

---

## [Editor Report · Decision Letter 1]

11 Aug 2021

Dear Dr Cassidy,

We are pleased to inform you that your manuscript 'The role of memory in non-genetic inheritance and its impact on cancer treatment resistance' has been provisionally accepted for publication in PLOS Computational Biology.

Best regards,

Dominik Wodarz

Associate Editor

PLOS Computational Biology

Natalia Komarova

Deputy Editor

PLOS Computational Biology

---

## [Editor Report · Acceptance letter]

26 Aug 2021

PCOMPBIOL-D-21-00385R1 

The role of memory in non-genetic inheritance and its impact on cancer treatment resistance

Dear Dr Cassidy,

I am pleased to inform you that your manuscript has been formally accepted for publication in PLOS Computational Biology. Your manuscript is now with our production department and you will be notified of the publication date in due course.

With kind regards,

Zsofi Zombor
